# Bibliometric Analysis on Research Trend of Accidental Falls in Older Adults by Using Citespace—Focused on Web of Science Core Collection (2010–2020)

**DOI:** 10.3390/ijerph18041663

**Published:** 2021-02-09

**Authors:** Boyuan Chen, Sohee Shin

**Affiliations:** 1School of Physical Education (Main Campus), Zhengzhou University, Zhengzhou 450001, China; cby@haust.edu.cn; 2School of Sport and Exercise Science, University of Ulsan, 93 Daehak-ro, Nam-gu, Ulsan 44610, Korea

**Keywords:** older adults, accidental falls, research hotspot, CiteSpace, knowledge domain visualization

## Abstract

The present study aimed to identify the trends in research on accidental falls in older adults over the last decade. The MeSH (Medical Subject Headings) and entry terms were applied in the Web of Science Core Collection. Relevant studies in English within articles or reviews on falls in older adults were included from 2010 to 2020. Moreover, CiteSpace 5.6.R5 (64-bit) was adopted for analysis with scientific measurements and visualization. Cooper Cyrus, Stephen R Lord, Minoru Yamada, Catherine Sherrington, and others have critically impacted the study of falls in older adults. Osteoporosis, dementia, sarcopenia, hypertension, osteosarcopenia, traumatic brain injury, frailty, depression, and fear of falling would be significantly correlated with falls in older adults. Multiple types of exercise can provide effective improvements in executive cognitive performance, gait performance, quality of life, and can also lower the rates of falls and fall-related fractures. Fall detection, hospitalization, classification, symptom, gender, and cost are the current research focus and development direction in research on falls in older adults. The prevention of falls in older adults is one of the most important public health issues in today’s aging society. Although lots of effects and research advancements had been taken, fall prevention still is uncharted territory for too many older adults. Service improvements can exploit the mentioned findings to formulate policies, and design and implement exercise programs for fall prevention.

## 1. Introduction

With aging, inactivity can lead to adverse and deep consequences, including health, economic, environmental, and social effects [1]. Many people are subject to multiple chronic diseases and drugs in their daily lives [2], which overall elevate the risk of falls for older adults. The annual incidence of falls for people aged over 65 is 30%–40%, and the incidence of falls for people aged over 80 is as high as 50% [3,4], which causes premature mortality, loss of independence, placement in assisted-living facilities, and death. A classification of fall risk factors has been proposed according to extrinsic and intrinsic factors. The extrinsic factors are related to surrounding space or environment-related, taking up to 30%–50% in most series [5], (e.g., tripping, slipping, walking on uneven surfaces, and inadequate illumination). The intrinsic or individual-related causes include advanced age, gait and balance impairment, concomitant chronic conditions (e.g., cardiovascular diseases, and sensory impairment), cognitive deficits, disorders of the central nervous system, severe osteoporosis with spontaneous fracture, and acute illness, drug side-effects, alcohol intake, anemia, hypothyroidism, unstable joints, and foot problems [6].

Falling always causes severe injuries, which is one of the costliest health conditions among older adults, imposing a heavy burden on the health care system [7]. An evidence-based fall prevention program can not only significantly reduce the incidence of fall-related injuries and medical costs but also improve the quality of life of older adults. As revealed from high-certainty evidence, exercise can prevent falls [8], which could reduce the rate of falls by 23% and down-regulate the number of people that experience one or more falls by 15% in community-dwelling older adults [9]. Furthermore, exercise-based interventions as a cost-effective treatment to prevent falls can benefit older adults substantially by improving their health, independence, and quality of life. Accordingly, regular screening for fall risk and care, and interventions, should be implemented in older adults.

Falls are usually multifactorial, effective prevention strategies are essential to reduce the public health burden on the increasing number of falls and fall-related mortality. Some countries recommend annual instability screening in people aged 65 or over, the geriatric specialists for coping with falls and other geriatric syndromes are increasingly demanded. To this end, it is necessary to conduct a comprehensive summary of the research about accidental falls in older adults over the last decade. An evidence-based fall prevention program can not only significantly reduce the incidence of fall-related injuries and medical costs, but also enhance the public’s awareness, and improve the quality of life of older adults. Therefore, articles or reviews in English were downloaded from the Web of Science Core Collection from 2010 to 2020, and Citespace 5.6.R5 (64-bit) (Chaomei Chen, Philadelphia, PA, USA) was used for visualization and interpretation to identify the status and focus of studies regarding falls in older adults, presenting the development direction for following falls studies in older adults.

## 2. Method

### 2.1. Data Selected

The input data of this study was found using a combination of the research results from the multiple topic search queries into the Web of Science Core Collection. This study employed the MeSH and entry terms singularly or in combination (34,899, see Appendix A). First, we ensured that the data being used was from 2010 to 2020.

The second placed stress on older adults and falls. One of the topic terms included “accidental fall*”, fall*, “fall*, accidental”, “fall and slip”, “slip and fall”. This query produced 171,659 records as Set #6. Besides, another topic term consists of aged, elderly, this query led to 1,705,687 records as Set #9. At last, we combined Set #6 and Set #9 together and got the final dataset, Set #10, containing 34,899 records.

Similar queries #1–#10 were employed here to retrieve bibliographic records on the common data sources for science mapping, including PubMed (14,025, see Appendix B), Embase (15,588, see Appendix C), Scopus (33,624, see Appendix D). Books, documents, and research grants, or other types of data sources may be required to be considered. However, this review is only limited to the records of types of articles or reviews in English in the Web of Science Core Collection.

All bibliographic information was downloaded and saved as plain text files for subsequent data processing and analysis. Subsequently, the data were imported into the Citespace and the duplicate data were deleted to prepare for the next step of visualization.

### 2.2. Data Analysis Method

Citespace refers to an information visualization tool extensively applied in the field of knowledge graphs [10]. Visualization tools were adopted to display and analyze the knowledge context of a certain domain, and the development process and structural relationship in this domain were suggested. Therefore, this review adopted CiteSpace 5.6.R5 (64-bit) to achieve visualization to gain insights into this field of accidental falls in older adults and discover the research frontier and knowledge base of the field in considerable data.

Notably, when the clustering function was started, the Modularity Q and the Mean Silhouette scores critically impacted visualization, representing an overall structural characteristic of the network. Overall, Q > 0.3 displayed an overall significant structure. If S > 0.5 or higher, the cluster was usually considered to be reasonable [11].

## 3. Result

### 3.1. Analysis Results and Visualization

#### 3.1.1. Basic Statistical Analysis

The number of papers published regarding falls in older adults was elevated from 2127 in 2010 to 4244 in 2020 (Figure 1). It is suggested that falls in older adults are attracting rising attention from researchers.

#### 3.1.2. Distribution of Journal Papers

Table 1 lists the top 10 journals that published the largest number of papers regarding falls in older adults from 2010 to 2020. PLoS One published about 609 papers, ranking the first. Overall, the specific subject scope comprises Geriatrics Gerontology, Medical General Internal, Public Environmental Occupational Health, Gerontology, Rehabilitation, Orthopedics, Clinical Neurology, Surgery, Neurosciences, Sport Sciences, and so on. In the listed top 10 journals, the highest impact factor is Age and Ageing, nearly 4.902.

#### 3.1.3. Co-Institution Analysis

We ran CiteSpace, generating a network as usual: 2010–2020, Slice length: 1 year; Node Select the node type: Institution, Top N = 20, choice Pathfinder and Pruning the merged network. Other parameters were the default settings. Also, the Co-institutions knowledge mapping was generated, in which N = 60, E = 67 (density was 0.0379).

Figure 2 indicates that the main research strengths were in universities. The University of Sydney has published the most papers and has conducted strong scientific research in the study on falls in older adults. Furthermore, the greatest number of bursts in the study was Harvard Medical School, reaching 59.04. The University of Oxford was the institution with the strongest centrality, reaching 8. The highest-ranked by Sigma was the University of Pittsburgh.

#### 3.1.4. Co-Author Analysis

By analyzing the author, the cooperative relationship with others could be investigated. We ran CiteSpace, generating a network as usual: 2010–2020, Slice length: 1 year; Node Select the node type: Author, Top N = 20, and choice Pathfinder and Pruning the merged network, other parameter settings were likely to institutions. This study found knowledge mapping of the co-author with N = 186, E = 186 (a density of 0.0108) (Figure 3).

Table 2 shows that Stephen R. Lord ranked first in the number of citations, with 175 citations. The most obvious Burst referred to the Minoru Yamada, reaching 12.13. The strongest centripetal force was Cooper Cyrus, displaying a centripetal force of 8. The highest Sigma (∑) was Catherine Sherrington, and the Sigma was 0.32.

The Web of Science was used (Table 3), and Cooper Cyrus’s H-index was 144. Professor Cyrus leads an internationally competitive program of research into the epidemiology of musculoskeletal disorders, most notably osteoporosis. Stephen R Lord has published over 600 papers in the areas of balance, gait, falls in older people, and is acknowledged as a leading international researcher in his field. His research primarily focuses on two themes: the identification of physiological risk factors for falls and the development and evaluation of fall prevention strategies. Minoru Yamada’s H-index was 28; cited 2501 times. Also, his research follows three main themes: the epidemiological study on sarcopenia and frailty; the effect of a care prevention program on healthy life expectancy; and the effect of physical activity on health outcomes in older adults. Catherine Sherrington’s H-index reached 52; cited 11,623 times. Currently, she is leading the Physical Activity, Ageing, and Disability Research Stream within the Institute, and is focused on health, aging falls, and rehabilitation.

### 3.2. Keyword Cluster Analysis

#### 3.2.1. Keyword Analysis

Keyword frequency analysis helps clarify the research trends on falls in older adults. Risk, balance, mortality, and prevalence were relatively high with frequencies of more than 2000 times, and prevention, gait, injury, women, exercise, hip fracture, community, quality of life, exercise, fracture, care, and management were relatively high with frequencies over 1000 times.

#### 3.2.2. Keyword Cluster Analysis

We ran CiteSpace, generating a network as usual: 2010–2020, Slice length: 1 year; Node Select the node type: Keyword; Top N = 100 and choice Pathfinder and Pruning the merged network. Given the co-occurrence of keywords, the nodes were revised, and the Log-likelihood (LLR) algorithm was adopted for clustering calculation. The visualization map obtained N = 147, E = 150 (density = 0.014), the Modularity Q score was 0.8423, the Mean Silhouette score was 0.6805, as presented in Figure 4.

There was a total of 20 clusters, mainly including 14 clusters, as listed in Table 4. Research topics regarding falls in older adults can be separated into two main topics. The first topic is risk factors that may cause accidental falls (e.g., #1 osteoporosis, #10 dementia, #13 fear of falling). The other one refers to intervention to prevent falls (e.g., #11 exercise, #12 vitamin D).

#### 3.2.3. Research Hot Spots and Path Analysis

A timeline visualization depicts clusters along a horizontal timeline. The main 14 clusters are presented in Figure 5. Each one can indicate the evolution of research in the field on falls in older adults from 2010 to 2020.

#### 3.2.4. Keywords Citation Bursts Analysis

Citation burst refers to keywords appearing suddenly in a short period or which usage frequency increases sharply. Overall, it reveals the evolution of the research topic in different periods, as listed in Table 5.

## 4. Discussion

### 4.1. Main Research Scholars’ Views

Age-related anatomic and functional changes in perception, neuromuscular function, and cognitive systems impair the control of balance and gait. Targeted training can improve muscle strength, balance, gait, mobility, while preventing falls in older adults [12], so fall prevention programs should be tailored to older adults’ level of physical well-being [13].

Postmenopausal women aged over 50 are at an increased risk of developing sarcopenia and osteoporosis. Accordingly, healthy lifestyle measures in women aged over 50 are essential for healthy aging [14]. Besides, the combination of optimal protein intake and exercise leads to a greater degree of benefit than either intervention alone.

Osteosarcopenia refers to a novel syndrome that often commonly appears in a frail subset of older adults. Combined with pharmacological, nutritional, and exercise-based interventions, it should enable a more comprehensive approach to mitigate osteosarcopenia in the future [15].

Exercise and fall prevention interventions should combine with special cultures and positively exploit the support from society, physicians, and families [16]. Health care professionals should routinely discuss fall prevention with older adults, provide evidence-based advice during consultations, and follow up with referrals [17]. In addition, dual-task training, cognitive-motor training, reactive step training, and multicomponent exercise programs can effectively improve executive cognitive, gait performance, and quality of life [18], as well as lower the rates of falls and fall-related fractures [19,20]. An environmental intervention perspective combined with adequate follow-up can successfully reduce community-dwelling older adults’ falls [21].

### 4.2. Main Clusters Analysis

#1 Osteoporosis. Osteoporosis is a silent disease until a fracture occurs, which has widely developed as a worldwide health problem for men and women aged over 50. Lumbar muscle strength and the presence of osteoporosis are endogenous factors of the risk of falls [22]. Compared with women without osteoporosis, women with postmenopausal osteoporosis had a history of one or more falls in the past year and were at a higher risk of recurrent falls so that at-risk populations should be identified through early diagnosis and treatment [22]. Balance training may significantly reduce the frequency of falls in osteoporosis patients [23]. Activities aimed at developing muscle strength, body balance, and improving intrinsic receptive sensation should be encouraged [24]. The potential consequences of severe osteoporosis can be mitigated by pharmacological therapies and the proper selection of modalities [25].

#2 Gait and #13 Fear of falling. These are common with advancing age. Decreased attention while walking is a significant risk factor for falls among community-dwelling older adults. Impairments in balance and gait are critical to older adults because they jeopardize the independence and contribute to the risk of falls and injuries [26]. A cut-off gait speed of 1.0 m/s can be a useful tool to identify individuals who are high-risk individuals and evaluate preventive interventions [27]. The number of medications was associated with a decrease in gait performance. Each additional medication up-regulates the risk of gait decline by 12% to 16% [28]. A history of falls in the previous year was a good predictor of the fear of falling, and fear of falling is an independent risk factor for falls in older adults. Falls Efficacy Scale-International (FES-I) and Tinetti’s Falls Efficacy Scale are reliable and valid to measure the fear of falling [29,30]. Whether exercise interventions reduce the fear of falling beyond the end of the intervention has been insufficiently evidenced [31].

#3 Trauma and #5 Mortality. Research on traumatic brain injury (TBI) has increased over the past two decades [32]. TBI is the main cause of emergency department visits in older adults, which is a significant part of the overall injury burden [33]. The major consequences of TBI are hip fractures and intracranial injury, which account for 46% of fatal falls in older adults [34]. Moreover, TBI arising from closed head trauma (CHT) significantly increases the risk of developing Alzheimer’s disease (AD), Parkinson’s disease (PD), and chronic traumatic encephalopathy (CTE) [35], and these would increase the risk of fall-related injuries in older adults. The incidence of TBI may continue to increase over time. Trauma patients with these risk factors may require higher professional health care levels and should be enrolled in a formally fall prevention program [36]. Moreover, trauma in older adults should be addressed from a public health vision with improved social service quality and prevention. Falls are a significant cause of mortality in older adults [37]. Unintentional falls continued to be a major cause of death (29%) in China [38]. The trend in mortality from falls was similarly observed increasing among US and European data [39,40,41,42]. The fall-related mortality in Japanese older adults aged 65–74 years showed a more rapid and continuous decreasing trend, but men over 75 years did not decrease [43].

#4 Aging and #11 Frailty. With the increasing older adult population, frailty is an important health care topic for people with geriatric syndromes. The effect of satisfaction with aging as a potential protective mechanism against fall results in reducing the risk for falls [44]. Frailty and pre-frailness are significantly associated with a higher risk of fracture, disability, and falls [45]. The future fall risk attributed to frailty was suggested to be higher in men than in women [46]. Accordingly, older adults should be evaluated for the possibility of geriatric syndromes to lower the risk of falls, fractures, or death.

#6 Exercise and #9 Vitamin D. Exercise programs reduce the rate of falls. An exercise program primarily involves balance and functional training [9], while a program includes multiple types of exercise (usually balance and functional exercises and resistance exercises) [47]; Otago exercise program, high-intensity interval training (HIIT), or virtual reality (VR) will have more significantly reduced the fall rate [48,49,50]. The interaction of exercise and various nutrients, especially protein and some multi-nutritional supplements, influenced muscle and bone health in older adults. Low levels of vitamin D have been associated with increased fall rates. However, no consistent conclusion has been reached for the relationship between vitamin D deficiency and these broader health outcomes [51], including daily oral doses of vitamin D [52,53,54]. Subsequent research should be conducted to determine the role of vitamin D in the relationship with falls in older adults.

One point that needs to be emphasized is that most countries have taken active interventions to prevent falls in older adults, significantly reducing the rate of falls in older adults. However, our society still lacks awareness of sarcopenia (Table 4 #9). The underlying mechanism of sarcopenia remains unclear, and no widely accepted definitions are suitable for use in research and clinical settings, and methodological challenges and debates are ongoing [55]. Sarcopenia has been associated with aging and older adults, but the development of sarcopenia now can also possibly occur earlier in life [56], so this study attempted to give a brief introduction to sarcopenia.

In 1989, Irwin Rosenberg proposed the theory of sarcopenia. In 2016, the ICD-10-MC Diagnosis Code officially identified sarcopenia as a muscle disease [57]. In 2010, the European Working Group on Sarcopenia in Older People (EWGSOP) developed practical clinical definitions and consensus diagnostic criteria [56] and updated the definition of myasthenia gravis by exploiting the last decade’s research and accumulated clinical evidence in 2018 [58]. The Asian Sarcopenia Working Group (AWGS) defined the diagnostic sarcopenia criteria by referencing Asian data in 2014 [59], while diagnostic procedures, protocols, and some metrics were revised in 2019 [60]. Both the Foundation for the National Institutes of Health (FNIH) and the International Working Group on Sarcopenia (IWGS) also have their definitions of sarcopenia. Thus, research based on different definitions may be misleading and difficult to interpret, such as cutoff point, diagnostic procedures, and so on.

Here, the definition of sarcopenia by the EWGSOP 2 is taken as an example. Sarcopenia is a progressive and generalized skeletal muscle disorder that is associated with an increased likelihood of adverse outcomes including falls, fractures, physical disability, and mortality [58]. Nutritional, inactivity, disease, iatrogenic may be the most frequent underlying causes of sarcopenia [61]. Specifically, sarcopenia is probable when low muscle strength is detected. A sarcopenia diagnosis is confirmed by the presence of low muscle quantity or quality. When low muscle strength, low muscle quantity/quality, and low physical performance are all detected, sarcopenia is considered severe. Subsequently, studies found that, when untreated, sarcopenia can bring a high personal, social, and economic burden [62]. For human health, sarcopenia can elevate the risk of falls and fractures [63], impairing the ability to perform activities of daily living [64], as well as raising the risk of hospitalization.

Lifestyle interventions, especially exercise and nutritional supplementation, prevail as mainstays of treatment. Subsequent research is required to investigate the potential long-term benefits of lifestyle interventions, nutritional supplements, or pharmacotherapy for sarcopenia. Moreover, several questions should be studied in-depth, including how to identify the high risk of sarcopenia early, what makes sarcopenia worse, which muscle indicators can be the most effective predictors of adverse outcomes, how we can optimally assess the muscle mass, how to determine effective critical value, which measurement tools are the most accurate, what interventions are available for sarcopenia, as well as which intervention should be the first choice.

#7 Epidemiology and #8 Validity. The incidence of falls and related complications increases with age. Furthermore, the epidemiology of falls in the incidence for women was higher than men. The rate of falls in community-dwelling adults is lower than in long-term care institutions. Most community-dwelling falls lead to about 5% fracture or hospitalization, and those in long-term institutions tend to more serious, with 10–25% of such falls resulting in fracture or laceration. Wrist fractures are common between the ages of 65 and 75, while hip fractures predominate after age 75 [5]. Wrist fractures usually result from falls forward or backward on an outstretched hand and hip fractures typically from falls to the side [65]. A lot of fall assessment tools have been developed and designed for different purposes over recent decades, most of them are targeted at assessing geriatric patients and have been available on reliability and validity [66]. But patient fall risk scales more focus on specific intrinsic and extrinsic factors, it could not fully assess a patient’s current fall risk status, which needs more patient-centered assessments and interventions [67].

#10 Dementia and #12 Depression. People with various levels of cognitive impairment can benefit from supervised multimodal exercise to improve physical function [68]. With the incidence of dementia growing globally, people with dementia are at a higher risk of falls and fall-related injuries, while there is still an argument about the exercise intervention for dementia patients [69]. There is little evidence about the effect of specific types of exercise on dementia risk [70]. More high-quality intervention studies should be conducted to inform evidence-practice initiatives. Depression is associated with the incidence of dementia, with a variety of possibly psychological or physiological mechanisms. Depression and falls are common and co-exist. Geriatric depression score (GDs) was used as a significant predictor of older adults from falls. Depression treatment should be incorporated in fall prevention programs for older adults at a high risk of increasing/multiple falls. Based on the existing state of knowledge, exercise (especially tai chi) and cognitive behavioral therapy should be considered to treat mild depression in older fallers [71].

### 4.3. Keywords Citation Bursts

According to Table 5, hypertension and postmenopausal women have attracted widespread attention at first. In postmenopausal women, due to insufficient estrogen, osteoporosis affects bone formation and increases the risk of falls. Women have caught the attention of scholars, and studies had proved that exercise training for postmenopausal women is an effective approach to improve fall or fracture [72]. The studies conducted between 2010–2015 on links between hypertension and falls are also of high significance. It is known that the increased risk of falls due to hypertension is related to the use of antihypertensive drugs, vascular sclerosis, and poor gait performance. Accordingly, nursing and intervention guidance should be strengthened to prevent patients with hypertension falls. With studies conducted in-depth, bone mineral density, postural balance, and body composition became research hotspots, and then gradually turned to fall prevention and prediction. Note that fall detection, classification, hospitalization, cost, and gender are receiving more attention.

A fall detection system by exploiting the Internet of Things can reduce the serious consequences of falls [73], which has made important progress in novel sensors, technologies, and algorithms [74]. However, there are two main challenges facing fall detection systems. One is to identify when a serious fall takes place, the other one refers to the lack of real data on falls to improve the research. Furthermore, how to apply laboratory data to real life, how to protect user privacy, and how to shift from detecting falls to predicting falls are recognized as the novel directions of development [75]. The classification of falls and the incidence of falls in different settings, socio-demographic determinants, international trends, and the measurement of fall outcomes, including the costs of falls and fall-related injuries, are the hot topics [76]. Costs generated by falls are expected to increase with the rapid expansion of the aging population. These costs fall into two parts. One is direct costs including health care costs (e.g., medications), while the other is treatment and consultations for rehabilitation, i.e., losses in societal productivity of activities for individuals and caregivers [77]. Occupational therapy had the effectiveness and cost-effectiveness in improving functional ability and decrease hospital readmission for older adults [78]. Risk factors for falls vary with gender [79]. Gender should be considered in the design of fall prevention strategies [80].

## 5. Conclusions

First, studies on falls in older adults have been increasingly conducted in the 21st century; the number of papers published every year is increasing. The mentioned findings suggest that when setting the selection criteria for Top N = 20, Cooper Cyrus, Stephen R Lord, Minoru Yamada, and Catherine Sherrington play an important role in the study of falls in older adults. The University of Sydney (Australia) has published the largest number of papers on falls in older adults, and the most obvious burst in the present study is Harvard Medical School (USA). The University of Oxford (USA) was the most central institution. The highest Sigma (∑) is The University of Pittsburgh (USA).

Second, Geriatrics Gerontology, Medical General Internal, Clinical Neurology, Clinical Neurology, Neurosciences, Orthopedics, Rehabilitation, Surgery, Sport Sciences, Public Environmental Occupational Health and Gerontology are considered the main research scopes involved in falls. The journals PloS One, Gait & Posture, and BMC Geriatrics were the top 3 journals regarding accidental falls in older adults.

Third, osteoporosis, dementia, sarcopenia, hypertension, traumatic brain injury, frailty, depression, fear of falling would be significantly correlated with falls in older adults. Nowadays, fall detection, hospitalization, classification, gender, and cost are the focus and direction of the development of falls in older adults.

Fourth, age-related changes in perception, neuromuscular, and cognitive systems interfere with the control of balance and gait. Fall prevention programs should be tailored to the older adult’s level of physical well-being. Targeted training can improve muscle strength, balance, gait, and mobility while preventing falls in older adults. A program consisting of multiple types of exercise, HIIT, or VR may more significantly reduce the fall rate of older adults than a single exercise intervention.

Falling is a serious issue concerned with the health of older adults, which affects the physical and mental health and quality of life for themselves and their families. Only one-third of all older adults who fell have sought medical assistance; one possible reason is the lack of public awareness about the importance of fall prevention [81]. This study introduces the last decade of research results on fall-related factors from different aspects such as physiology, pathology, psychology, environment, and sports science expounds on the latest developments in this aspect of research and relevant experts’ opinions are summarized, enabling more people to gain comprehensive insights into falls of older adults to prevent or reduce older adults from fall-related injuries. Moreover, a good peer view is presented for the study using scientific methods to find good methods to prevent, treat or reduce the risk of falls. There are some limitations to the study. For instance, the selected papers were only included in the Web of Science Core Collection, and searches are not selected in PubMed, Scopus, or other databases. Besides, the literature contains papers in English only; the status of research on falls in older adults in other language nations is not possible to determine. Lastly, CiteSpace analysis is biased towards quantitative analysis. In subsequent studies, the qualitative research method of the interview method should be adopted to remedy the defects of quantitative research. Though further research is needed, this preliminary result may give a new horizon for fall prevention.

## Figures and Tables

**Figure 1 ijerph-18-01663-f001:**
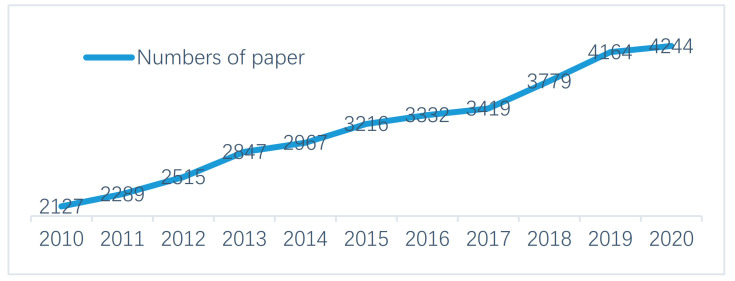
Papers regarding accidental falls in older adults (2010–2020).

**Figure 2 ijerph-18-01663-f002:**
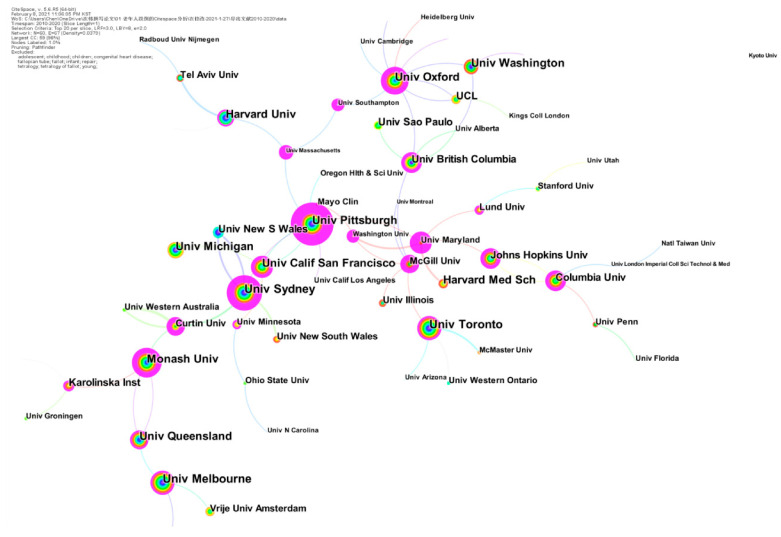
Co-institutions’ network (2010–2020). The color of the circle represents when the article was published. The larger the node diameter, the more papers institutions have published. The thicker the line between the nodes, the closer the two institutions work together.

**Figure 3 ijerph-18-01663-f003:**
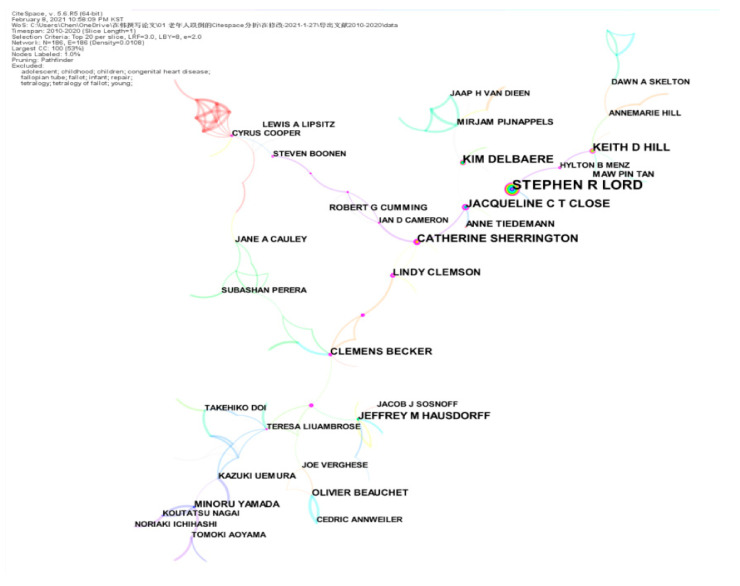
Co-authorship network (2010–2020). The color of the line represents the time the co-authors worked together. The larger the node diameter, the more papers the author has collaborated to publish. The thicker the line between the nodes, the closer the cooperation between the two authors.

**Figure 4 ijerph-18-01663-f004:**
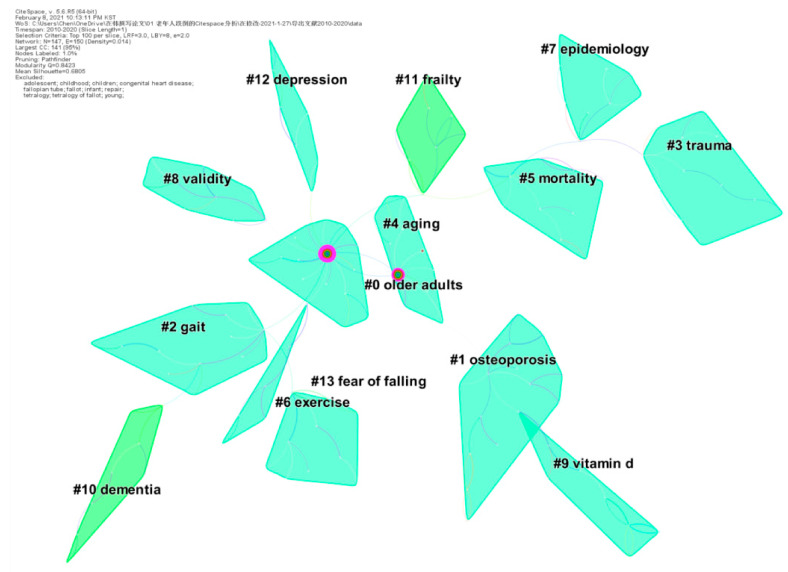
A landscape view of keyword cluster analysis generated by Top N = 100 per slice from 2010 to 2020. (LRF = 3, LBY = 8, and e = 2.0).

**Figure 5 ijerph-18-01663-f005:**
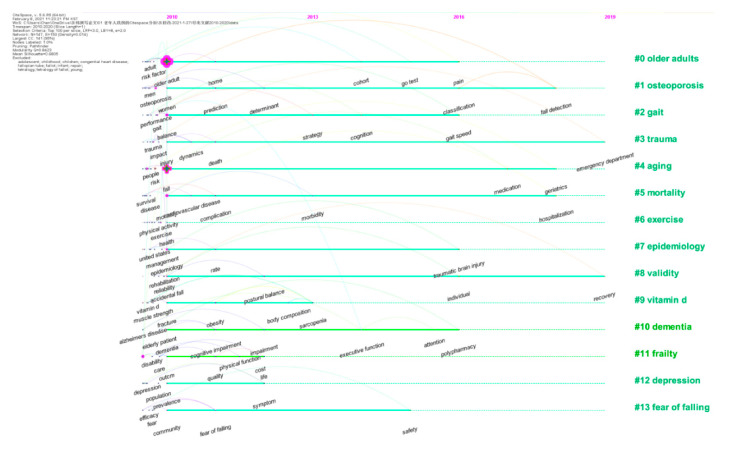
A timeline of the 14 largest clusters in accidental falls in older adults (2010–2020).

**Table 1 ijerph-18-01663-t001:** Top 10 journal published analysis (2010–2020).

No.	Journal Title	IF	Amount	Country	Research Area
1	PLoS One	2.740	609	USA	Science & Technology-Other Topics (Q2)
2	Gait & Posture	2.349	425	Ireland	Neuroscience & Neurology (Q3) Orthopedics (Q2) Sport Sciences (Q2)
3	BMC Geriatrics	3.077	366	England	Geriatrics & Gerontology (Q2)
4	Journal of the American Geriatrics Society	4.180	321	USA	Geriatrics & Gerontology (Q1)
5	Archives of Gerontology and Geriatrics	2.128	272	Ireland	Geriatrics & Gerontology (Q3)
6	Osteoporosis International	3.864	271	England	Endocrinology & Metabolism (Q2)
7	Aging Clinical and Experimental Research	2.697	264	Italy	Geriatrics & Gerontology (Q3)
8	BMJ Open	2.496	219	England	General & Internal Medicine (Q2)
9	Journal of the American Medical Directors Association	4.367	197	USA	Geriatrics & Gerontology (Q1)
10	Age and Ageing	4.902	196	England	Geriatrics & Gerontology (Q1)

**Table 2 ijerph-18-01663-t002:** Author rank in different conditions.

No.	Co-Authorship Papers	Burst	Centrality	Sigma
1	Stephen R Lord	Minoru Yamada	Cooper Cyrus	Catherine Sherrington
2	Kim Delbaere	Koutatsu Nagai	Mirjam Pijnappels	Clemens Becker
3	Catherine Sherrington	Kazuki Uemura	Teresa Liu-Ambrose	Jacqueline C T Close
4	Keith D Hill	Anne-Marie Hill	Keith D Hill	Jorunn L Helbostad
5	Jacqueline CT Close	Noriaki Ichihashi	Jeffrey M Hausdorff	Lindy Clemson

Centrality represents the degree of nodes that are part of the path that connects any pair of nodes in the network. Burst refers to the specific time during which a sudden change in frequency occurs. Sigma measures a combination of structural and temporal characteristics of nodes.

**Table 3 ijerph-18-01663-t003:** Researchers’ academic information.

Researcher	H-Index	Sum of Cited	Research Areas
Cooper Cyrus	144	106,019	Osteoporosis & Osteoarthritis & Epidemiology
Stephen R Lord	93	91,427	Falls in Older People
Jeffrey M Hausdorff	76	28,247	Gait & Neurodynamic
Catherine Sherrington	52	11,620	Health & Exercise & falls & Ageing & Rehabilitation
Keith D Hill	44	6778	Falls prevention & Exercise & Rehabilitation
Teresa Liu-Ambrose	41	5789	Fall prevention & Healthy aging
Jacqueline C T Close	40	6746	Gait & Gerontology & Geriatric Assessment
Kim Delbaere	38	4792	Ageing & Accidental falls & Fear of falling & Cognitive function
Clemens Becker	36	4654	Falls & Exercise & Rehabilitation
Jorunn L Helbostad	31	3476	Movement disorders and falls at old age
Minoru Yamada	28	2501	Gerontology & Rehabilitation
Mirjam Pijnappels	28	2787	The effects of aging on neuromuscular and cognitive aspects of mobility
Noriaki Ichihashi	26	2192	Rehabilitation & Physical therapy
Lindy Clemson	25	2972	Ageing & Occupational Therapy
Kazuki Uemura	23	1802	Rehabilitation & Welfare engineering
Koutatsu Nagai	17	816	Gerontology & Physical Therapy
Anne-Marie Hill	16	1043	Falls prevention & Patient education

**Table 4 ijerph-18-01663-t004:** Subjects of cluster analysis (2010–2020).

Clusters	Silhouette	Size	Log-Likelihood (LLR)
#0 older adults	15	1	Risk factor, mobility, pain, prevention, quality of life
#1 osteoporosis	14	0.968	Fall detection, bone mineral density, classification, machine learning, fracture, wearable sensors
#2 gait	13	0.985	Balance, walking, postural control, variability, parkinsons disease, gait analysis
#3 trauma	12	0.966	Injury, impact, emergency department, frailty, suicide
#4 aging	11	1	Hip fracture, falls, vitamin d supplementation, prescription, Romberg test
#5 mortality	11	1	Blood pressure, survival, surgery, hypertension, morbidity, disease
#6 exercise	10	0.971	Physical activity, health, intervention, randomized controlled trial, fitness, social participation
#7 epidemiology	9	0.956	Traumatic brain injury, management, trend, diagnosis, rehabilitation
#8 validity	9	0.967	Reliability, women health, knee pain, lower extremity, inertial sensors, practice guidelines
#9 vitamin d	8	1	Sarcopenia, fractures, physical performance, obesity, muscle strength, osteosarcopenia,
#10 dementia	8	0.966	Cognitive impairment, polypharmacy, motoric cognitive risk syndrome, attention, long-term care
#11 frailty	8	0.952	Care, quality, patient, comprehensive geriatric assessment, disability,
#12 depression	7	0.96	Prevalence, population, mental health, sleep quality, behavior, anxiety,
#13 fear of falling	6	0.918	Efficacy, safety, exercise, nurses, fear, physical activity monitoring

**Table 5 ijerph-18-01663-t005:** 38 Keywords with the strongest citation bursts (2010–2020).

Keywords	Year	Strength	Begin	End	2010–2020
vitamin d	2010	25.7615	2010	2013	▃▃▃▃ ▂▂▂▂▂▂▂
infection	2010	34.6606	2010	2012	▃▃▃ ▂▂▂▂▂▂▂▂
double blind	2010	22.0649	2010	2011	▃▃ ▂▂▂▂▂▂▂▂▂
growth	2010	4.5875	2010	2012	▃▃▃ ▂▂▂▂▂▂▂▂
history	2010	36.0171	2010	2013	▃▃▃▃ ▂▂▂▂▂▂▂
community	2010	10.2497	2010	2011	▃▃ ▂▂▂▂▂▂▂▂▂
guideline	2010	30.1278	2010	2013	▃▃▃▃ ▂▂▂▂▂▂▂
hypertension	2010	28.7127	2010	2015	▃▃▃▃▃▃ ▂▂▂▂▂
randomized controlled trial	2010	16.4364	2010	2011	▃▃ ▂▂▂▂▂▂▂▂▂
older people	2010	55.1187	2010	2014	▃▃▃▃▃ ▂▂▂▂▂▂
postmenopausal women	2010	16.4984	2010	2012	▃▃▃ ▂▂▂▂▂▂▂▂
bone mineral density	2010	2.6016	2010	2011	▃▃ ▂▂▂▂▂▂▂▂▂
older women	2010	31.0928	2010	2013	▃▃▃▃ ▂▂▂▂▂▂▂
dynamics	2010	26.5791	2011	2012	▂ ▃▃ ▂▂▂▂▂▂▂▂
home	2010	48.2979	2011	2014	▂ ▃▃▃▃ ▂▂▂▂▂▂
follow up	2010	16.818	2012	2017	▂▂ ▃▃▃▃▃▃ ▂▂▂
controlled trial	2010	23.878	2012	2013	▂▂ ▃▃ ▂▂▂▂▂▂▂
postural balance	2010	15.4265	2012	2017	▂▂ ▃▃▃▃▃▃ ▂▂▂
rate	2010	25.3309	2013	2014	▂▂▂ ▃▃ ▂▂▂▂▂▂
body composition	2010	32.8776	2013	2014	▂▂▂ ▃▃ ▂▂▂▂▂▂
prediction	2010	2.6749	2013	2014	▂▂▂ ▃▃ ▂▂▂▂▂▂
fall prevention	2010	25.6649	2013	2014	▂▂▂ ▃▃ ▂▂▂▂▂▂
consequence	2010	27.3192	2014	2015	▂▂▂▂ ▃▃ ▂▂▂▂▂
cohort	2010	43.2065	2014	2018	▂▂▂▂ ▃▃▃▃▃ ▂▂
safety	2010	28.2582	2015	2016	▂▂▂▂▂ ▃▃ ▂▂▂▂
attention	2010	33.7397	2016	2017	▂▂▂▂▂▂ ▃▃ ▂▂▂
experience	2010	13.7373	2016	2018	▂▂▂▂▂▂ ▃▃▃ ▂▂
pain	2010	28.5537	2016	2018	▂▂▂▂▂▂ ▃▃▃ ▂▂
individual	2010	33.4865	2016	2020	▂▂▂▂▂▂ ▃▃▃▃▃
trial	2010	9.7152	2016	2017	▂▂▂▂▂▂ ▃▃ ▂▂▂
symptom	2010	32.2689	2016	2020	▂▂▂▂▂▂ ▃▃▃▃▃
gait speed	2010	23.6379	2016	2017	▂▂▂▂▂▂ ▃▃ ▂▂▂
life	2010	16.1569	2017	2018	▂▂▂▂▂▂▂ ▃▃ ▂▂
classification	2010	37.0551	2018	2020	▂▂▂▂▂▂▂▂ ▃▃▃
fall detection	2010	48.4538	2018	2020	▂▂▂▂▂▂▂▂ ▃▃▃
cost	2010	16.6557	2018	2020	▂▂▂▂▂▂▂▂ ▃▃▃
hospitalization	2010	37.8174	2018	2020	▂▂▂▂▂▂▂▂ ▃▃▃
gender	2010	27.1045	2018	2020	▂▂▂▂▂▂▂▂ ▃▃▃

▃▃: shows which period the citation burst is the strongest. For instance, the postural balance has the longest period of burst from 2012 to 2017.

## Data Availability

Not applicable.

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
