# Peer review of "Bibliometric Analysis on Research Trend of Accidental Falls in Older Adults by Using Citespace—Focused on Web of Science Core Collection (2010–2020)"

_ijerph, 2021, doi:10.3390/ijerph18041663_

Round 1

Reviewer 1 Report

Phrases are detected with: ERROR in appointment !!! (lines: 67, 75, 105, 112, 137, 142, 150, 193)

The meaning of what is expressed in lines 233 and 234 is not understood

The study time (January 1, 2009 and September 23, 2020) is irregular. It is recommended to justify the deadlines used

Author Response

We are grateful to reviewers for the critical comments and useful suggestions that have helped us to improve our paper considerably.

As indicated in the following responses, we have incorporated all their comments into the revised version of our paper.

Best wishes.

Reviewer 2 Report

The study conducted a bibliometric analysis of studies on accidental falls in older adults to track  the research focus and development direction from the Web of Science Core Collection database. CiteSpace was used to perform analysis and visualization of co-institution, co-author and keyword cluster. The study addressed a relevant research topic and it meets the scope of the journal. However, adjustments and clarifications are needed in order to improve the rationale of the study, methodological decisions, and results. Please see the specific comments.

  1. The authors need to address detailed background information and relevantstudies about accidental falls in elderly, not only falls in the introduction section.
  2. The database Web of Science (Core Collection) was searched for eligible papers from 2009 to September 23, 2020. The author should claim the reason for choosing this time interval.
  3. In the method section, the MeSH terms and keywords can not accurately reflect the topic, leading to the uncorrelated articles screened in this study. The detailed data retrieval strategies and inclusion criteria should be summarized.
  4. The analysisof the keyword in the related articles is important in the bibliometrics. Reference co-citation analysis (RCA) is also the core indices. I suggest the authors add this part.
  5. Line 223, According to Table 1, in the early stages of the study on accidental falls in the older adults. Table 1 describes the journal analysis, which should be clarified.

Author Response

(The authors gave the same response as above.)

Reviewer 3 Report

INTRODUCTION

In general, the introduction is too short. More detailed information on falls in older adults should be included to understand the problem and the purpose of the study.

METHODS

The data presented in this section are clear and concise. However, I believe that the authors can add relevant information to improve the research design.

  • Add information about the study design.
  • The eligibility criteria for the selection of articles are not specified. Please add.
  • Lines 46-48: You must specify the combinations used to search for articles.

RESULTS

  • In this section the expression appears repeatedly: “Error! Ref-105 reference source not found”. Please, restate.
  • The types of study of the selected articles are not specified in any section of the results. This information would be relevant for a higher quality of this study.

DISCUSSION

  • In this section, the expression also appears repeatedly: “Error! Ref-105 reference source not found”. Please, restate.

CONCLUSION

  • In general, it would be necessary to add the clinical impact of the data collected in this work. It appears that the authors did the study well, but did not answer any clinical questions with their work.

REFERENCES     

  • Please review this section. It must have a homogeneity.

Author Response

(The authors gave the same response as above.)

Reviewer 4 Report

General comments:

Congratulations for this work, I add comments to make your paper as complete and better.

In all manuscript, there are grammatical errors of "." and words that remain attached to "." You must separate the words and carefully review some grammar mistakes.

Lines (32, 57, 67, 88, 164, 174, 207, 213, 232 and 234).

INTRODUCTION

the Introduction is too short, you should mention or add aspects of what influences people to produce a fall (extrinsic and intrinsic factors). It would also be important to define how general physical activity causes improvements in older adults and positively reduces the risk of falls and falls risk factors, as well as how all this influences the reduction of healthcare costs. Also, later in the discussion, name different exercise programs that substantially improve the risk of falls.

METHODS:

One of the strengths of this study is the 10-year search for articles related to such an important topic in aging and the health of the older people. But it has a weakness, it is due to the fact that this search is only carried out in a database (WOS). Why haven't other databases such as Scopus or PubMed been used? I know that you contemplate it in limitations, but it is important to know your choice.

RESULTS:

In Table 2, it would be important to add for each one of the authors,  H-index, number of cites and principal topic related with falls. The topic would be to attach the most important variables studied by each of the authors in relation to falls.

CONCLUSION

In the final section, you comment about the limitations of your study, but you must add an increase in your strengths from said study, talk about the importance of falls, how it reduces costs and how with this increase in papers all this can be reduced, in addition to giving you more value to your so specific search and what future research could be based on your contribution to the research.

DISCUSSION

  In the first part, you name different types of exercise programs. In this part it would be important to add how novel programs such as HIIT in older people cause and reduce the risk of falls, as is the case of the study by Jiménez-García et al. Doi: 10.1123/japa.2018-0190 and title: Risk of Falls in Healthy Older Adults: Benefits of High-Intensity Interval Training Using Lower Body Suspension Exercises. Furthermore, you can add other similar studies with novel programs exercises such "multicomponent exercise program" and fall risk.

Author Response

(The authors gave the same response as above.)

Round 2

Reviewer 2 Report

Line 551, except for the falls and road injuries, TBI also occurs in a wide range of contact sports (boxing, rugby, and wrestling). It is estimated that approximately 1.6 to 3.8 million sports related TBI occur every year, accounting for roughly 15% of all high school sport-related trauma reported. It may be interesting to discuss or cite the following papers: Qi et al. Int. J. Environ. Res. Public Health, 2020, 17, 5411; VanItallie et al. Metabolism, 2019, 100, 153943.

Author Response

We are grateful to reviewers for the critical comments and useful suggestions that have helped us to improve our paper considerably. As indicated in the following responses, we have incorporated the comment into the revised version of our paper.

Reviewer 4 Report

The manuscript has improved significally. You have improved or added all comments that I have realized to improve this manuscript. Good job.

Author Response

Thank you so much.
I was so nervous because I am a beginner. But it is my luck to meet a reviewer like you.
It is precise because of your guidance that I received a lot of meaningful help and made corrections in my research. At the same time, I also feel your patience with me. You are like a teacher guiding your students(Chinese call it "en-shi",恩师).
I will also summarize the experience I have gained this time. In future research, I will definitely work harder.
I think this is not only helpful for my own future research but also I will pass on your rigorous, responsible, knowledgeable, and diligent academic spirit.
Thank you again for your help.